# Alpha-Synuclein-Specific Regulatory T Cells Ameliorate Parkinson’s Disease Progression in Mice

**DOI:** 10.3390/ijms242015237

**Published:** 2023-10-16

**Authors:** Seon-Young Park, HyeJin Yang, Soyoung Kim, Juwon Yang, Hyemin Go, Hyunsu Bae

**Affiliations:** 1Department of Science in Korean Medicine, College of Korean Medicine, Graduate School, Kyung Hee University, 26 Kyungheedae-ro, Dongdaemun-gu, Seoul 02447, Republic of Korea; psys12@naver.com (S.-Y.P.); emilly86@naver.com (H.Y.); samanda0@nate.com (S.K.); 2Department of Korean Medicine, College of Korean Medicine, Graduate School, Kyung Hee University, 26 Kyungheedae-ro, Dongdaemun-gu, Seoul 02447, Republic of Korea; jwyang@khu.ac.kr (J.Y.); comrp1521@hanmail.net (H.G.)

**Keywords:** Parkinson’s disease, regulatory T cell, alpha-synuclein

## Abstract

Parkinson’s disease (PD) is a long-term neurodegenerative disease characterized by dopaminergic neuronal loss and the aggregation of alpha-synuclein (α-syn) in the brain. Cell therapy using regulatory T cells (Tregs) has therapeutic potential on PD progression in a mouse model; however, several challenges were associated with its applications. Here, we propose a strategy for α-syn specific Treg expansion (α-syn Treg). We presented α-syn to T cells via dendritic cells. This method increased the mobility of Tregs towards the site of abundant α-syn in vitro (*p* < 0.01; α-syn Tregs versus polyclonal Tregs (poly Tregs)) and in vivo. Consequently, α-syn Tregs showed noteworthy neuroprotective effects against motor function deficits (*p* < 0.05, *p* < 0.01; α-syn Tregs versus poly Tregs), dopaminergic neuronal loss (*p* < 0.001; α-syn Tregs versus poly Tregs), and α-syn accumulation (*p* < 0.05; α-syn Tregs versus poly Tregs) in MPTP-induced PD mice. Furthermore, the adoptive transfer of α-syn Tregs exerted immunosuppressive effects on activated microglia, especially pro-inflammatory microglia, in PD mice. Our findings suggest that α-syn presentation may provide a significant improvement in neuroprotective activities of Tregs and suggest the effective clinical application of Treg therapy in PD.

## 1. Introduction

Parkinson’s disease (PD) is a progressive neurodegenerative disorder that leads to motor deficits, including tremors, rigidity, and postural instability [1]. The current therapy for Parkinson’s is aimed only at relieving the disease symptoms. None of the prescribed drugs, such as dopamine agonists and levodopa, have been proven disease-modifying [2]. Additionally, there are many side effects, such as drug-related motor fluctuations, on–off effects, and dyskinesia [3].

A characteristic feature of PD is the presence of soluble or insoluble aggregates of alpha-synuclein (α-syn) in the brain. Though α-syn is a synaptic protein present in the brain, it accompanies neurotoxicity and ultimately leads to neuronal dysfunction [4]. Aggregated α-syn is phagocytosed by microglia and induces microglial activation with ROS production. Activated microglia are found in specific brain regions in patients with PD and are considered influential in the degeneration of dopaminergic neurons by amplifying neuronal inflammation [5,6]. The 1-methyl-4-phenyl-1,2,3,6-tetrahydropyridine (MPTP) model is the most widely used experimental model for studying PD in mice. MPTP causes α-syn accumulation in the substantia nigra (SN) and PD-like motor impairment [7,8]. In the MPTP model, as in patients with PD, microglial activation is induced before the dopaminergic neuronal loss and is implicated in PD pathology. Therefore, activated microglia are a promising therapeutic target for PD [9].

CD4^+^CD25^+^ regulatory T cells (Tregs) play a crucial role in controlling immune balance in the central nervous system (CNS). However, their suppressive activity is dysregulated during inflammation, aggravating the damage caused by autoreactive T effector cells [10,11]. In line with this, Tregs have been considered an attractive therapeutic modality that induces neuroprotective activity [12]. Adoptive transfer of Tregs has been attempted as a potential treatment for neurodegenerative diseases such as Alzheimer’s disease (AD), amyotrophic lateral sclerosis (ALS), and even PD [13,14,15]. However, these studies have some limitations. A major problem with polyclonal Treg (poly Treg) therapy is the suppression of off-target immune responses, which may increase susceptibility to opportunistic infections or inhibit antitumor activity. As a solution to this problem, the development of antigen-specific Tregs has been attempted. It has been reported that antigen-specific Tregs are more effective and safer than polyclonal Tregs in a mouse model of non-obese diabetes and graft-versus-host disease (GVHD) [16,17,18,19]. Therefore, α-syn, a PD-specific antigen, could increase the effectiveness of Treg therapy targeting PD.

The aim of the current study is to establish the effects of α-syn presentation on the ex vivo expansion of Tregs (α-syn Tregs). Thus, we adoptively transferred expanded Tregs into an MPTP-induced mouse model of PD and assessed the disease pathology. The advantage of α-syn presentation showed the enhancement of the neuroprotective effects of Tregs on motor function, neuronal loss, and pro-inflammatory microglial activation. In conclusion, our findings provide insight into the clinical application of Treg therapy which has great potential for PD.

## 2. Results

### 2.1. α-syn Tregs Maintain the Suppressive Phenotype during Ex Vivo Expansion

To prepare antigen-specific suppressive Tregs, CD4+ T cells were co-cultured with poly- or α-syn-presenting DCs for four days. After 4 days, poly- or α-syn-specific CD4^+^CD25^+^ Tregs were isolated and expanded for 14 days (Figure 1a). The expression of CD62L in Tregs was analyzed using flow cytometry (Figure 1b). At the start (day 0), there were no differences between poly and α-syn Tregs. However, on day 14, a higher expression of CD62L was observed in α-syn Tregs (Figure 1c). After expansion, mRNAs were extracted from Tregs, and the expression of Treg markers, such as FoxP3, GATA3, IL-10, and Amphiregulin (Areg), was analyzed (Figure 1d). The expression of these markers was increased in α-syn Tregs. These results suggest that α-syn Tregs maintain the immunosuppressive phenotype during ex vivo expansion.

### 2.2. Antigen Presentation Increases the Mobility of Tregs toward Disease Site

To determine whether antigen presentation affected TCR Vβ usage, the TCR repertoire was assessed using the Mouse Vβ TCR Screening Panel (Figure 2a). The usage of some Vβ, particularly Vβ 8.1/2, 8.3, and 13, increased slightly after antigen presentation.

It is presumed that antigen-specific Tregs are more efficient than poly Tregs because they migrate towards the specific area of the brain where the antigen exists [20]. Therefore, we performed an in vitro migration assay to assess the mobility of Tregs towards α-syn (Figure 2b). After 4 h of seeding Tregs, approximately 50% of α-syn Tregs moved to the bottom where α-syn existed, whereas less than 10% of poly Tregs moved. To determine whether these in vitro results were reproduced in vivo, we produced poly and α-syn Tregs expressing Thy1.1 from Thy1.1^+^ mice (Figure 2c). These Tregs were adoptively transferred to Thy1.2^+^ MPTP-intoxicated mice. Their distributions were detected with Thy1.1 expression in CD4^+^7AAD^−^ cells. The results showed that α-syn Tregs accounted for a higher proportion in the brain, while there were no differences between the groups in other tissues such as lung, spleen, kidney, and lymph node. These results support that α-syn presentation improves the mobility of Tregs to α-syn-rich environments in vitro as well as in vivo.

### 2.3. α-syn Tregs Improve Motor Function in MPTP-Induced Parkinson’s Disease Mice

To investigate the effect of Tregs on MPTP-induced neurodegeneration, 5 × 10^5^ Tregs were adoptively transferred into MPTP-treated mice (12 mg/kg) (Figure 3a). A pole test was carried out to examine PD-related motor deficits on day 6, and mice were sacrificed on day 7. MPTP treatment causes deficits in motor function resembling those observed in human PD [21]. As expected, MPTP treatment significantly extended the time taken to turn downward compared to the wild type (WT) (Figure 3b). This time was significantly shortened in α-syn Tregs but not in poly Tregs. In particular, α-syn Tregs significantly improved motor function compared with poly Tregs in both indices. These results suggest that α-syn Tregs significantly improve motor function.

### 2.4. α-syn Tregs Reduce Parkinson’s Disease Pathology in MPTP-Induced Parkinson’s Disease Mice

PD is characterized by the loss of dopaminergic neurons in the SN [22]. To confirm the loss of dopaminergic neurons, TH was assessed in the SN (Figure 4a). MPTP treatment significantly reduced the number of TH-positive dopaminergic neurons compared with the control. α-syn Tregs significantly inhibited the loss of dopaminergic neurons compared with both MPTP and poly Tregs.

To investigate the effects of Tregs on the upregulation of α-syn in the MPTP-induced mouse model of PD, SNs were stained with α-syn (Figure 4b). The α-syn intensity in the SN was increased in MPTP-intoxicated mice; α-syn Tregs decreased the accumulation of α-syn in the SN compared to MPTP and poly Tregs. These results suggest that antigen presentation improves the effects of Tregs on PD pathology.

### 2.5. α-syn Tregs Modulate Microglial Polarization in MPTP-Induced Parkinson’s Disease Mice

Activated microglia secrete M1-associated pro-inflammatory cytokines that aggravate neurodegeneration in the PD brain, while M2 microglia participate in tissue repair by producing anti-inflammatory cytokines [23]. To assess the activation of pro-inflammatory M1 microglia, SNs were immunostained with Iba1 as a marker of activated microglia and NOS2 as a marker of pro-inflammatory microglia (Figure 5A). The increased intensities of NOS2 and Iba1 in MPTP-treated mice were significantly decreased only by the adoptive transfer of α-syn Tregs. To further confirm the effects of Tregs on microglial polarization, the mRNA expression of M1 and M2 microglial markers was evaluated with quantitative real-time PCR (Figure 5B,C). The relative expression of NOS2, an M1 phenotypic marker, was significantly reduced only by the α-syn Treg transfer. Interestingly, the expression levels of the pro-inflammatory cytokines TNF-α, IL-1β, and IL-6 were significantly decreased with any Tregs. However, the adoptive transfer of Tregs has no significant effect on the relative mRNA levels of Arg1 and IL-4, an M2 phenotypic marker. Additionally, the mRNA levels of FoxP3 and IL-10 were significantly increased in the α-syn Treg-transferred mice brains (Figure 5D). Collectively, these data show that α-syn Tregs exert a better therapeutic effect than polyclonal Tregs.

## 3. Discussion

Our present study proposes α-syn presentation as a method for enhancing the therapeutic effects of Tregs in PD. To prepare Tregs, CD4^+^CD25^+^ Tregs were isolated and expanded ex vivo after α-syn presentation via DCs. The α-syn Tregs showed the following characteristics. First, the expression of suppressive markers was steadily maintained. Second, these Tregs moved more frequently towards α-syn both in vitro and in vivo. Finally, Tregs showed better neuroprotective effects on PD pathology with improved motor deficits, dopaminergic neuronal loss, and α-syn accumulation.

Tregs tightly regulate immune balance in the CNS as immune suppressors. In neurodegenerative diseases, Treg dysfunction leads to neuroinflammation. Many studies have focused on the importance of Tregs in the pathology of neurodegenerative diseases [10,24]. Tregs are significantly reduced or show poor suppressive activity in mild AD, ALS, and multiple sclerosis (MS) [25,26,27]. In line with this, Treg therapy has been attempted, and its neuroprotective effects have been demonstrated in neurodegenerative diseases, including AD, ALS, and PD [13,14,15]. In patients and mouse models with PD, CD4^+^ and CD8^+^ T cells were demonstrated to infiltrate the brain [28,29]. Other studies have reported that CD3^+^ T cells were accumulated in the brains of AD and MS patients [30,31]. These studies suggest the possibility of infiltration of adoptively transferred cells into the brain with neurodegenerative diseases and its potential as a therapeutic strategy for these diseases targeting adaptive immunity.

For efficient Treg therapy, it is important to diminish the number of Tregs to be administrated. Previously, Ashley et al. demonstrated that adoptively transfer of higher than 3.5 × 10^6^ Tregs was necessary to elicit sufficient neuroprotective effects in a behavior test and TH immunoreactivity MPTP-mice model [13]. Similarly, in AD and ALS models, the neuroprotective effects of Tregs have been shown by transferring 1 × 10^6^ or more Tregs which were cultured with stimulation for 4 days [14,15]. To obtain such an amount of Tregs in humans, ex vivo expansion is essential for patients and many protocols for the ex vivo expansion of Tregs have been developed [18,32,33]. In this study, we adoptively transferred only 5 × 10^5^ ex vivo expanded Tregs to MPTP-intoxicated mice and demonstrated that they could show sufficient neuroprotective effects with α-syn presentation. However, since the Tregs should be isolated from peripheral blood mononuclear cells (PBMC) in humans for clinical Treg therapy, there is a need for adequate modifications with antigen presentation and expansion.

For the application of Treg therapy to actual patients, it is also desirable to increase their suppressive potential. Several studies suggest the importance of CD62L expression in Tregs for clinical manipulation in GVHD [34,35]. Highly expressed CD62L is rapidly lost after antigen experience [36]. However, we maintain the expression of CD62L using bee venom phospholipase A2 (bvPLA2), a Treg inducer maintaining CD62L expression during α-syn presentation [37]. Finally, we also examined the expression of Treg-specific genes. Areg is a critical factor for Treg suppressive function in vivo and in vitro [38]. Tregs produce Areg and IL-10 for tissue repair in damaged tissues, regulated by GATA3 [39]. These results demonstrate that our method of preparation of α-syn Treg enhances the efficiency of Treg therapy. A previous study has reported that in PD, peripheral DCs could migrate across the blood–brain barrier and uptake α-syn aggregates in the brain and then proceed to the cervical lymph nodes in which they enhance the MHC-II expression and α-syn presentation to activate antigen-specific T cells [40,41]. Ng et al. found that treatment of α-syn with monocyte-derived DCs stimulated robust inflammatory α-syn specific CD4+ T cell responses. They also found that both endo-lysosomal and autophagic pathways are associated with α-syn, and treatment of DCs with α-syn appeared to downregulate the expression of Rab and autophagosomal proteins, suggesting the possibility that α-syn regulated its interaction with antigen processing components which limited its degradation and allowed the activation of T cells [40]. Alam et al. confirmed that the expression of co-stimulatory (CD80, CD83, and CD86) and MHC (HLA-ABC) molecules upregulated in DCs after treating α-syn to DCs. It has also been confirmed that α-syn can stimulate phenotypic and functional maturation of both DCs, endowing them with increased antigen-presenting capacity [42]. Like these studies, additional studies about antigen presentation of DCs need to be added in future experiments, although we presented α-syn to T cells via DCs and found that antigen presentation increases Tregs migration toward the brain region where the antigen exists (Figure 2).

Microglia, the primary immune cells in the CNS, play crucial roles in orchestrating brain inflammation [43]. Microglia may perform pathogenic or neuroprotective functions, depending on their phenotype. M1 microglia express NOS2 as a phenotypic marker and secrete pro-inflammatory cytokines such as IL-1β, IL-6, and TNF-α. M2 microglia, which express Arg1, are associated with tissue repair by producing IL-4 and IL-10 [23]. Activated microglia are involved in the pathogenesis of certain neurological diseases such as PD [44]. In the brains of patients with PD, an increased level of activated microglia is representative, and it causes neurodegeneration. In animal models, activation of microglia and secretion of M1-associated pro-inflammatory cytokines are induced by MPTP [23]. Additionally, α-syn acts as a stimulus that activates M1 microglia in the PD brain [45]. Therefore, targeting the balance of impaired pro- and anti-inflammatory microglia can be a strategy for disease modification [46]. Tregs ameliorate neuroinflammatory injury by regulating microglial polarization. In addition, loss of Tregs induced pro-inflammatory microglial activation in ICH and stroke models [47,48]. Indeed, studies attempting Treg therapy in neurodegenerative diseases have commonly observed the inhibition of microglial inflammation after Treg transfer [13,14,15]. Our findings are consistent with those of previous studies showing that Tregs inhibit neuroinflammation by modulating microglial polarization. Since Th1 and Th17 cells contribute to neuroinflammation by accompanying proinflammatory cytokines to induce neurotoxicity by microglia in PD [49], we additionally measured the expression level of IFN-g and IL-17 with qPCR. However, there were no significant effects of Tregs.

The current data showed that α-syn presentation improved the neuroprotective effects of Tregs in a PD mouse model by ameliorating neurodegeneration, α-syn accumulation, and microglial inflammation. Our data suggest that α-syn presentation may provide a significant improvement in the neuroprotective activities of Tregs, indicating a potential therapeutic value. Ultimately, our strategy may provide a promising avenue for immunotherapy with clinical relevance and cost-effectiveness for treating PD. However, there are certain limitations. It remains unclear how α-syn Tregs improved motor function, and unlike α-syn Tregs, why poly Tregs failed to do so. It also remains unclear how distinct α-syn specific Tregs may reduce inflammation and neurodegeneration. In this regard, further studies are required for a better understanding of the mechanisms of Treg therapy. Resolving this issue will enable the development of a more efficient protocol for Treg expansion.

The implication that α-syn presentation improved the suppressive potential and mobility of Tregs towards the disease site in a mouse model of PD is clear. These improvements increase the therapeutic effects of Tregs on the amelioration of PD progression by modulating microglial polarization (Figure 6).

## 4. Materials and Methods

### 4.1. Instrumentation

The instruments used in the experiment are listed as follows (Table 1).

### 4.2. Reagents

Alpha-synuclein (α-syn) was purchased from Prospec (cat #PRO-393). Granulocyte-macrophage colony-stimulating factor was purchased from R&D systems (Minneapolis, MN, USA) (cat #415-ML-010-CF). CD11c (cat #130-052-001) and CD4^+^CD25^+^ Regulatory T Cell Isolation Kit (cat #130-091-041) were purchased from Miltenyi Biotec Inc. (Auburn, CA, USA). 1-Methyl-4-phenyl-1,2,3,6-tetrahydropyridine (MPTP)-hydrochloride (HCl) was obtained from Sigma-Aldrich (St. Louis, MO, USA) (cat #M0896). PE-Cy7-CD4 (cat #25-0041-82), APC-CD62L (cat #17-0621-82), PE-Cy5-CD4 (cat #15-0041-82), and 7AAD (cat #00-6993-50) were obtained from Invitrogen (Waltham, MA, USA); the APC-Cy7-CD25 (cat #557658), mouse Vβ TCR Screening Panel (cat #557004), PE-cy7-CD4 (cat #552775), and BV421-Thy1.1 (cat #740044) were from BD Bioscience. To detect dopaminergic neurons, primary antibodies directed against tyrosine hydroxylase (TH, cat #P40101-150) and an avidin-biotin complex kit (Vectastain ABC kit, cat #PK-6102) were purchased from Pel-Freez Clinical System (Rogers, AR, USA) and Vector Laboratories (Newark, CA, USA), respectively. For immunofluorescence, the primary antibodies against ionized calcium-binding adaptor molecule 1 (Iba1 cat #ab178846) were from Abcam (Cambridge, UK); the primary antibodies against nitric oxide synthase 2 (NOS2 cat #sc-7271) and α-syn were from Santa Cruz Biotechnology (Dallas, TX, USA). Alexa 488- or 594-conjugated IgG secondary antibodies were from Invitrogen.

### 4.3. Animals

Male C57BL/6 mice and Thy1.1 (B6.PL-Thy1<a>/CyJ) mice were obtained from Jackson Laboratory (Bar Harbor, ME, USA). The mice were maintained on a 12 h light/dark cycle and under temperature-controlled conditions, with food and water ad libitum. All experiments were performed in accordance with approved animal protocols and guidelines established by Kyung Hee University (KHUASP(SE)-22-148 and 20-240). 

### 4.4. Cell Preparation

To present the Parkinson’s disease-specific antigen to Tregs via dendritic cells (DCs), α-syn (Prospec, East Brunswick, NJ, USA) was aggregated with incubation in a shaker at 37 °C for 7 days. To obtain bone marrow (BM)-DCs, BM-leukocytes were isolated from the femurs and tibiae of mice and resuspended in a medium containing 20 ng/mL granulocyte-macrophage colony-stimulating factor (R&D Systems, Minneapolis, MN, USA) [50]. After 7 days, CD11c^+^ DCs were isolated using CD11c MicroBeads (Miltenyi Biotec Inc., Auburn, CA, USA) and seeded at a density of 2 × 10^5^/mL in 96-well U-bottom plates. For antigen presentation, the samples were treated with (for α-syn Tregs) or without (for poly Treg) 2 μg/mL aggregated α-syn for 24 h. CD4 T cells were isolated from splenocytes using CD4 (L3T4) microbeads (Miltenyi Biotec Inc.). The cells were added to the DC at a ratio of 10:1 (CD4 T: DC) with 0.4 μg/mL bvPLA2 (Sigma-Aldrich, St. Louis, MO, USA). Four days after CD4 T cells-DC co-culture, CD4^+^CD25^+^ T cells (Tregs) were isolated using magnetic-activated cell sorting, according to the manufacturer’s protocol (CD4^+^CD25^+^ Regulatory T Cell Isolation Kit; Miltenyi Biotec Inc.). The CD4^+^CD25^+^ T cells were stimulated using the Treg Expansion Kit (Miltenyi Biotec Inc.) for 14 days.

### 4.5. Treg Migration Assay

Treg migration was assayed using 24-well Transwell chambers with 5 μm pores. A total of 10^5^ cells in 100 μL of media (RPMI 1640 with 0.5% BSA) were placed in the upper chambers. Aggregated α-syn, diluted in 600 μL media, was placed in the lower wells, and the chambers were incubated at 37 °C. After 3 and 4 h, the migrated cells present in the bottom wells were counted using a LUNA-II automated cell counter (Logos Biosystems, Anyang, Republic of Korea).

### 4.6. Animal Experiment

For MPTP (1-methyl-4-phenyl-1,2,3,6-tetrahydropyridine) intoxication, seven-week-old male mice received four i.p. injections of MPTP-HCl (12 mg/kg free base in saline; Sigma-Aldrich) at 2 h intervals as previously described [13]. A wild-type mouse, which did not receive the injections of MPTP-HCl, was prepared for the control group. Twelve hours after the last MPTP injection, the MPTP-intoxicated mice were randomly divided and received adoptive transfers of 5 × 10^5^ Tregs (N = 8–10, total N = 37). The checklist for the guidelines for animal experiments is uploaded as a Appendix A.

### 4.7. Trafficking

For trafficking of adoptively transferred Tregs, poly and α-syn Tregs were prepared from Thy 1.1-mice as described in the cell preparation section. MPTP-treated mice (Thy 1.2^+^) were divided randomly into four groups and received an adoptive transfer of 1 × 10^6^ Thy 1.1^+^ Tregs. After 7 days, the mice were euthanized, and the inguinal lymph nodes, spleen, blood, lung, kidney, liver, and brain were harvested. T cells were enriched with 30–70% Percoll (Cytiva, Marlborough, MA, USA) density gradient centrifugation and debris removal solution (Miltenyi) from the lung, kidney, liver, and brain.

### 4.8. Flow Cytometry

For flow cytometry, cells were washed with BD FACS Stain buffer (BD Bioscience, San Jose, CA, USA) and stained with fluorescently labeled antibodies for 30 min at 4 °C in the dark. The following antibodies were used: PE-Cy7-CD4 (Invitrogen, Waltham, MA, USA), APC-CD62L (Invitrogen), and APC-Cy7-CD25 (BD Pharmingen, San Diego, CA, USA) for the Treg phenotype, PE-Cy5-CD4 (Invitrogen) and mouse Vβ TCR Screening Panel (BD Pharmingen) for the T cell receptor (TCR) repertoire, and PE-cy7-CD4, BV421-Thy1.1 (BD OptiBuild, San Jose, CA, USA), and 7AAD (Invitrogen) for Treg trafficking. All data were acquired using FACSLyric™ (BD Biosciences) and analyzed using the FACSuite software v1.2 (BD Biosciences).

### 4.9. Pole Test

Six days after the adoptive transfer of Tregs, a pole test was performed to determine forelimb and hindlimb motor coordination and balance. Briefly, the mice were placed on top of a gauze-banded wooden pole (50 cm in length and 0.8 cm in diameter) facing upward. The animals were allowed to climb down to the base of the pole. The time taken to turn completely downward and the total time taken for the mouse to reach the floor (time to down) were recorded. The maximum cut-off time to stop was 30 s.

### 4.10. Tissue Processing and Immunohistochemistry

Mice were anesthetized with isoflurane (Forane solution; ChoongWae Pharma, Seoul, Republic of Korea) and transcardially perfused with phosphate-buffered saline (PBS). Brains were dissected and divided in half. The brain halves were postfixed in 4% paraformaldehyde for 18 h at 4 °C, transferred to 30% sucrose solution, and subsequently frozen. Tissues were serially cut on a cryostat into 30-μm-thick coronal sections using a cryomicrotome (HM525 NX; Thermo Fisher Sientific, Inc., Waltham, MA, USA).

To detect dopaminergic neurons, primary antibodies were directed against tyrosine hydroxylase (TH; 1:2000, Pel-Freez Clinical System, Rogers, AR, USA). The sections were washed with PBS, incubated with the appropriate biotinylated secondary antibody, and processed using an avidin-biotin complex kit (Vectastain ABC kit; Vector Laboratories, Newark, CA, USA) for 1 h at 25 °C. The reaction product was visualized with 0.05% diaminobenzidine-HCl and 0.003% hydrogen peroxide in 0.1 M phosphate buffer. The labeled tissue sections were subsequently mounted and analyzed under a bright-field microscope (Nikon, Tokyo, Japan). An unbiased stereological estimation of the total number of TH-positive dopaminergic neurons in the SN was performed using the optical fractionator method on an Olympus computer-assisted stereological toolbox system version 2.1.4. (Olympus, Tokyo, Japan) as previously described [51]. The sections used for counting covered the entire SN, from the rostral tip of the pars compacta to the caudal end of the pars reticulata.

For immunofluorescence, brain sections were incubated with 50% formic acid for 10 min at 25 °C and heated with 10 mM sodium citrate buffer (pH 6.0) for epitope retrieval. After washing with cold PBS, nonspecific binding was reduced by blocking the sections with 5% BSA in 0.2% Triton X-100 in TBS for 30 min at 25 °C. The sections were incubated with antibodies against ionized calcium-binding adaptor molecule 1 (Iba1; 1:1000, Abcam, Cambridge, UK), nitric oxide synthase 2 (NOS2; 1:500, Santa Cruz Biotechnology, Dallas, TX, USA), and α-syn (1:100, Santa Cruz Biotechnology) O/N at 4 °C. Brain sections were washed with TBSTr, incubated for 2 h at 25 °C with Alexa 488- or 594-conjugated IgG secondary antibodies, and then counterstained with DAPI. The tissues were examined using an LSM 800 confocal laser-scanning microscope (Carl Zeiss, Oberkochen, Germany). The staining intensity was quantified by measuring the integral density of the region of interest from monochromatic images using the ImageJ software v1.52a. The percentage of staining intensity was calculated relative to the MPTP group and multiplied by 100.

### 4.11. RNA Extraction and Quantitative Real-Time PCR (qRT-PCR)

Total RNA was isolated from Tregs or unfixed brain halves using the easy-BLUE RNA extraction kit (iNtRON Biotechnology, Seongnam, Republic of Korea), and cDNA was synthesized using Cyclescript reverse transcriptase (Bioneer, Daejeon, Republic of Korea). Samples were prepared for real-time PCR using the SensiFAST SYBR no-Rox kit (Bioline, Randolph, MA, USA). The cycling conditions were: 1 cycle at 95 °C for 30 s, 49 cycles at 95 °C for 10 s, 55 °C for 30 s, followed by a melting curve at 9 5 °C for 10 s, 50 °C for 5 s, and then a gradual increase until 95 °C was reached. The base sequences of the primers were as follows: β-actin, forward: 5′-GTG CTA TGT TGC TCT AGA CTT CG-3′ and reverse: 5′-ATG CCA CAG GAT TCC ATA CC-3′; FoxP3: forward: 5′-CTG CTC CTC CTA TTC CCG TAA C-3′ and reverse: 5′-AGC TAG AGG CTT TGC CTT CG-3′; GATA3: forward: 5′-GAA GGC ATC CAG ACC CGA AAC-3′ and reverse: 5′-ACC CAT GGC GGT GAC CAT GC-3′; IL-10: forward: 5′-CAG CCG GGA AGA CAA TAA CTG-3′ and reverse: 5′-CCG CAG CTC TAG GAG CAT GT-3′; Areg: forward: 5′-ACT GTG CAT GCC ATT GCC TA-3′ and reverse: 5′-ACT GGG CAT CTG GAA CCA TC-3′; NOS2: forward: 5′-AGG ACA TCC TGC GGC AGC-3′ and reverse: 5′-GCT TTA ACC CCT CCT GTA-3′; TNF-α: forward: 5′-GGC AGG TTC TGT CCC TTT CAC-3′ and reverse: 5′-TTC TGT GCT CAT GGT GTC TTT TCT-3′; IL-1β: forward: 5′-AAG CCT CGT GCT GTC GGA CC-3′ and reverse: 5′-TGA GGC CCA AGG CCA CAG G-3′; IL-6: forward: 5′-TTC CAT CCA GTT GCC TTC TTG-3′ and reverse: 5′-GGG AGT GGT ATC CTC TGT GAA GTC-3′; Arg1: forward: 5′-CTC CAA GCC AAA GTC CTT AGA G-3′ and reverse: 5′-AGG AGC TGT CAT TAG GGA CAT C-3′; IL-4: forward: 5′-ATC CTG CTC TTC TTT CTC GAA TGT-3′ and reverse: 5′-GCC GAT GAT CTC TCT CAA GTG ATT-3′.

### 4.12. Statistical Analysis

All data were analyzed using GraphPad Prism 5.01 (GraphPad Software Inc., San Diego, CA, USA). The data are presented as the mean and standard error of the mean (SEM), where indicated. The statistical significance of each variable was evaluated with a two-tailed *t*-test for single comparison and one-way analysis of variance (ANOVA), followed by Tukey’s multiple comparison test for multiple comparisons. Multiple comparisons within groups were analyzed using a two-way ANOVA, followed by Bonferroni post-hoc tests. All experiments were performed in a blind manner and were repeated independently under identical conditions. Statistical significance was set at *p* < 0.05.

## Figures and Tables

**Figure 1 ijms-24-15237-f001:**
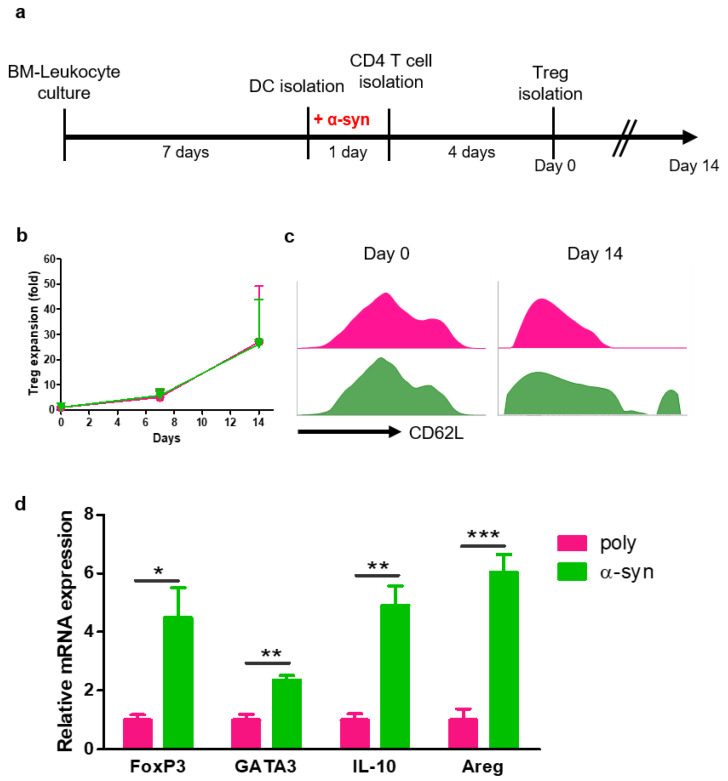
α-syn Tregs have a more suppressive phenotype. (**a**) Schematic diagram of Treg preparation and expansion. (**b**,**c**) On days 0, 7, and 14 after Treg isolation, the phenotypes of Tregs were analyzed using flow cytometry. (**d**) The relative mRNA expressions of FoxP3, GATA3, IL-10, and Areg in expanded Tregs were analyzed. Data are presented as mean ± SEM. Statistical analyses were conducted with two-tailed *t*-test; * *p* < 0.05, ** *p* < 0.01, and *** *p* < 0.001. *n* = 3–4.

**Figure 2 ijms-24-15237-f002:**
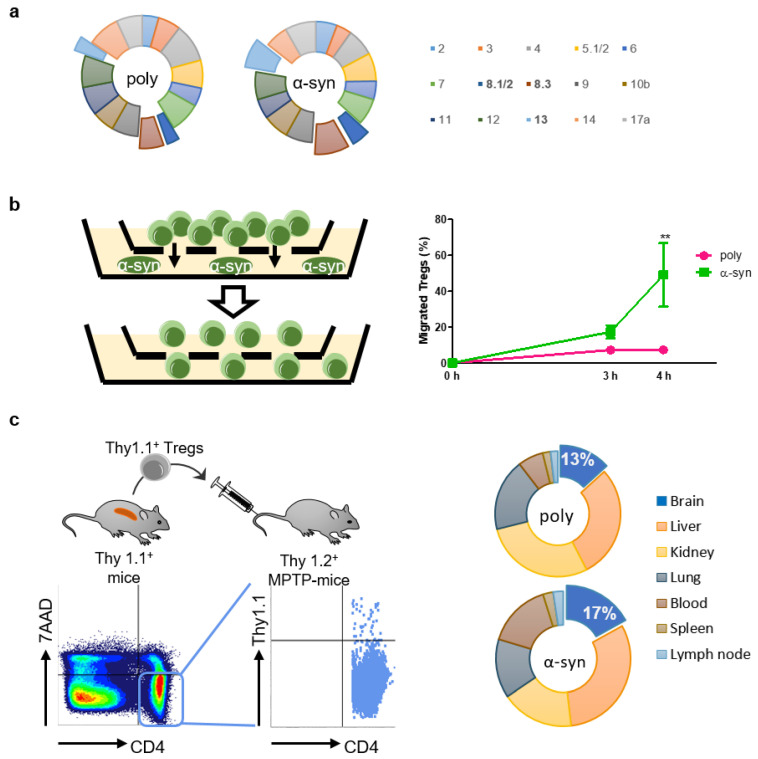
α-syn Tregs move more to site with α-syn. (**a**) TCR repertoire was assessed using Mouse Vb TCR Screening Panel. (**b**) Tregs and α-syn were seeded in the upper and bottom chambers of transwell, respectively. Migration of Tregs was assessed after 3 and 4 h. (**c**) Thy1.1^+^ Tregs were isolated from splenocytes of Thy1.1^+^ mice. Thy1.1^+^ Tregs were adoptively transferred to MPTP-intoxicated mice and detected in the lymph node, spleen, blood, lung, kidney, liver, and brain after 7 days. Data are presented as mean ± SEM. Statistical analyses were conducted with two-way ANOVA; ** *p* < 0.01. *n* = 3–4.

**Figure 3 ijms-24-15237-f003:**
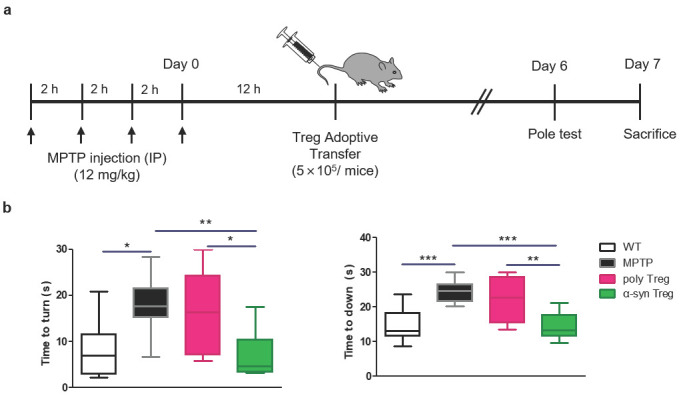
α-syn Tregs moderate motor dysfunction of MPTP-induced mice model of PD. (**a**) Schematic diagram of Treg transfer in MPTP-induced PD mice. (**b**) Time to turn downward and descend the pole was measured by pole test on day 6. Data are presented as mean ± SEM. Statistical analyses were conducted with one-way ANOVA; * *p* < 0.05, ** *p* < 0.01, and *** *p* < 0.001. *n* = 8–10.

**Figure 4 ijms-24-15237-f004:**
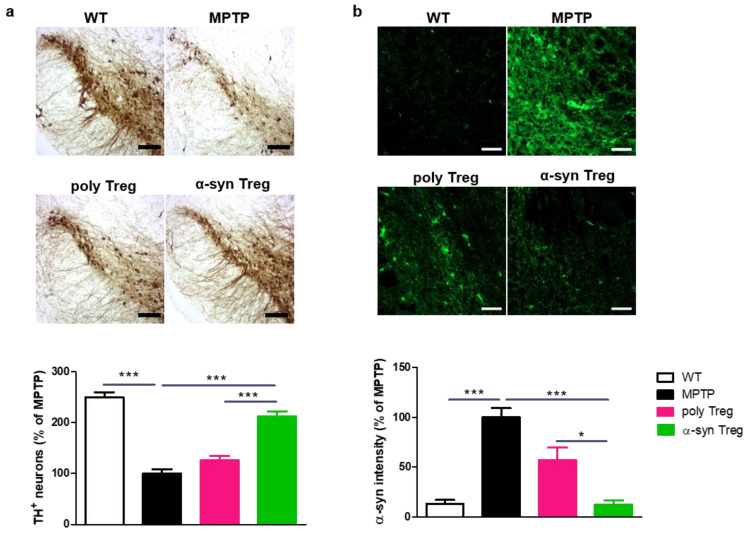
Immunohistochemical staining of TH and α-syn in the SN of MPTP-induced mice model of PD. (**a**) Immunohistochemistry was performed for expression of TH, a dopaminergic neuron marker, in SN of MPTP-induced mice model of PD. (**b**) To measure α-syn accumulation, brain sections were stained with α-syn. The intensity was calculated and normalized with MPTP. Data are presented as mean ± SEM. Statistical analyses were conducted with one-way ANOVA; * *p* < 0.05, and *** *p* < 0.001. *n* = 4–6. Scale bars: 50 μm.

**Figure 5 ijms-24-15237-f005:**
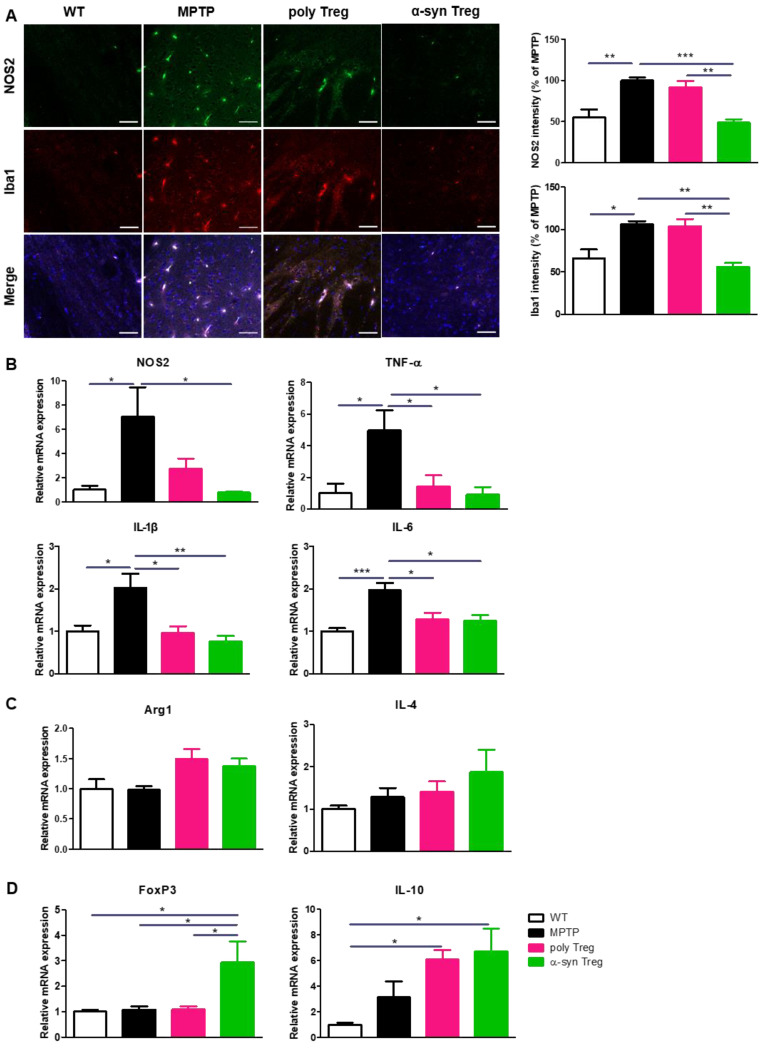
α-syn Tregs modulate microglial polarization. (**A**) The expression of NOS2 and Iba1 were detected in the SN. (**B**–**D**) The relative mRNA expressions of pro-inflammatory markers (NOS2, TNF-α, IL-1β, and IL-6), anti-inflammatory markers (Arg1, IL-4), and Treg markers (IL-10 and FoxP3) were analyzed in the brain. Data are presented as mean ± SEM. Statistical analyses were conducted with one-way ANOVA; * *p* < 0.05, ** *p* < 0.01, and *** *p* < 0.001. *n* = 4–6. Scale bars: 50 μm.

**Figure 6 ijms-24-15237-f006:**
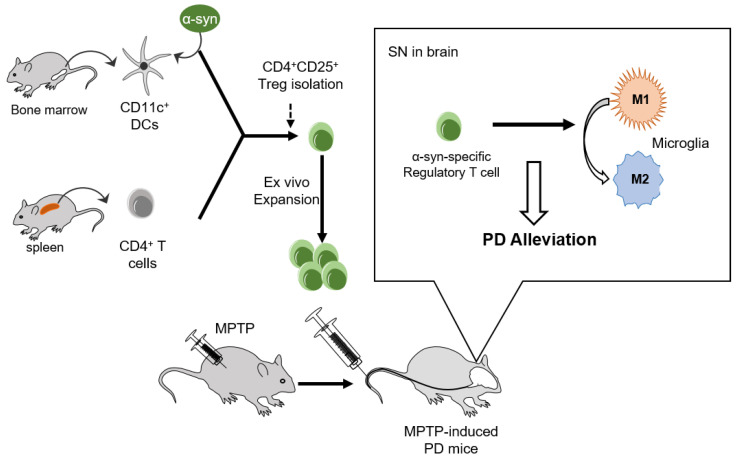
Adoptive transfer of α-syn-specific Tregs alleviated PD in MPTP-intoxicated mice. CD11C^+^ DCs and CD4^+^ T cells were isolated from the bone marrow and spleen, respectively. CD4^+^CD25^+^ Tregs were isolated and expanded ex vivo after α-syn presentation. Adoptive transfer of α-syn specific Tregs inhibited pro-inflammatory M1 microglial activation and alleviated PD pathologies such as motor dysfunction and dopaminergic neuronal loss in MPTP-induced PD mice.

**Table 1 ijms-24-15237-t001:** Instruments used in the study.

Instrumentation	Manufacturing Company/City, Country
LUNA-II automated cell counter	Logos Biosystems/Anyang, Republic of Korea
FACSLyric™ flow cytometer	BD Biosciences/San Jose, CA, USA
Cryomicrotome	Thermo Fisher Scientific/Waltham, MA, USA
Bright-field microscope	Nikon/Tokyo, Japan
LSM 800 confocal laser-scanning microscope	Carl Zeiss/Oberkochen, Germany
T100™ thermal cycler	BIORAD/Hercules, CA, USA
CFX connect Real-time PCR system	BIORAD/Hercules, CA, USA

## Data Availability

All data generated or analyzed during this study are included in this article.

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
