# Peer review of "Alpha-Synuclein-Specific Regulatory T Cells Ameliorate Parkinson’s Disease Progression in Mice"

_ijms, 2023, doi:10.3390/ijms242015237_

Round 1

Reviewer 1 Report

Kindly include the statistical finding inside the abstract.

Animal study must adhare arrive guideline. Please submit ARRIVE Checklist as supplementary file. https://arriveguidelines.org/

Please explain the problem statement inside introduction.

Kindly include the heading of instrumentation and provide the list with source.

Kindly include the heading chemicals and provide a list of chemical with source.

please provide a separate heading of strength and limitation and explain with 3-4 paragraph

good

Author Response

First of all, thank you for your interest in our paper, entitled on ‘Alpha-synuclein-specific regulatory T cells ameliorate Parkinson’s disease progression in mice’ to International Journal of Molecular Sciences. 
We did our best to modify it according to reviewer’s comments. 
Please see the attachment.

Reviewer 2 Report

Here the authors tested α-synuclein (α-syn ) specific Tregs  in neuroprotective effects against motor function deficits, dopaminergic neuronal loss, and α-syn accumulation in MPTP-induced mouse model. They showed that adoptive transfer of α-syn Tregs exerted immunosuppressive effects on activated microglia, especially pro-inflammatory microglia, in MPTP mice. Overall, their data suggest that α-syn presentation may provide a significant improvement in neuroprotective activities of Tregs, indicating a potential therapeutic value. This study is very straight forward. But it remains unclear how α-syn Tregs improved motor function but irrelevant Tregs failed to do so. Antigen presentation assays for α-syn Tregs should be shown. There are also concerns that polyclonal T cells reactive to multiple α-syn peptides (whole α-syn was used) may attenuate the disease. Since α-syn reactive T cells are mostly pathogenic in MPTP mouse model, it remains unclear whether Th1/Th17 cells are attenuated or dysregulated by Tregs. The manuscript also lacks mechanistic studies on how distinct α-syn peptide-reactive Tregs may reduce inflammation and neurodegeneration.

Minor comments:

Line 387, “All authors have read and agreed to the published version of the manuscript” this is not published.

Line 3958, “All data generated or analyzed during this study are included in this published article.”  Again, this is not published.

Some figures are difficult to read/interpret (Figure 5).

n/a

Author Response

(The authors gave the same response as above.)

Round 2

Reviewer 2 Report

The authors failed to address the comments raised by this reviewer. The responses are marginal.

Minor to moderate editing is needed.

Author Response

First of all, thank you for your interest in our paper, entitled on ‘Alpha-synuclein-specific regulatory T cells ameliorate Parkinson’s disease progression in mice’ to International Journal of Molecular Sciences. 
We did our best to modify it according to the reviewer’s comments.

Round 3

Reviewer 2 Report

The authors did not address the comments raised by this reviewer.

Minor editing is needed